

# Carbon Monitor Power - Simulators (CMP-SIM v1.0) across countries: a data-driven approach to simulate daily power demand

Léna Gurriaran[1, 2], Yannig Goude[3], Katsumasa Tanaka[1,4], Biqing Zhu[1,5], Zhu Deng[6], Xuanren Song[5], Philippe Ciais[1]

[1]Laboratoire des Sciences du Climat et de l'Environnement (LSCE), IPSL, CEA/CNRS/UVSQ, Université Paris-Saclay, Gif-sur-Yvette, France
[2]Atos, River Ouest, 95877 Bezons Cedex, France
[3]EDF Lab, 7 Boulevard Gaspard Monge, 91120 Palaiseau
[4]Earth System Division, National Institute for Environmental Studies (NIES), Tsukuba, Japan
[5]Department of Earth System Science, Tsinghua University, Beijing, 100084, China
[6]Alibaba Cloud, Hangzhou, Zhejiang, 310030, China

*Correspondence to*: Léna Gurriaran (lena.gurriaran@lsce.ipsl.fr; lena.gurriaran@atos.net)

**Abstract.** The impact of climate change on power demand has become increasingly significant, with changes in temperature, relative humidity, and other climate variables affecting cooling and heating demand for households

and industries. Accurately predicting power demand is crucial for energy system planning and management. It is also crucial to understand the evolution of power demand to estimate the amount of $CO_2$ emissions released into the atmosphere, allowing stakeholders to make informed plans to reduce emissions and adapt to the impacts of climate change. Artificial intelligence techniques have been used to investigate energy demand-side responses to external factors at various scales in recent years. However, few have explored the impact of climate and weather

variability on power demand. This study proposes a data-driven approach to model daily power demand provided by the Carbon Monitor Power project by combining climate variables and human activity indices as predictive features. Our investigation spans the years 2020 to 2022 and focuses on eight countries or groups of countries selected to represent different climates and economies, accounting for over 70 % of global power consumption. These countries include Australia, Brazil, China, the European Union (EU), India, Russia, South Africa, and the

United States. We assessed various machine-learning regressors to simulate daily power demand at the national scale. For countries within the EU, we extended the analysis to one group of countries. We evaluated the models based on key evaluating metrics: coefficient of determination ($R^2$), Mean Absolute Error (MAE), Root Mean Squared Error (RMSE), and Median Absolute Error (MedAE). We also used the models to identify the most influential variables that impact power demand and apprehend their relationship with it. Our findings provide

insight into variations in important predictive features among countries, along with the role played by distinct



climate variables and indicators of the level of economic activity, such as weekends and working days, vacations and holidays, and the influence of COVID-19.

# 1 Introduction

Climate significantly impacts power demand (Lucon et al., 2014; Isaac and van Vuuren, 2009), as the changes in
temperature, relative humidity, and precipitation patterns affect the cooling and heating demand of households and industries (Mukherjee et al., 2019). Globally, climate change is expected to increase total power demand under low latitudes and decrease under temperate and high latitudes because of warmer winters (Van Ruijven et al., 2019). However, there remain large uncertainties in how climate change will affect power demand (Deroubaix et al., 2021; Romitti and Sue Wing, 2022; Yalew et al., 2020) due to complexities associated with
understanding the precise effects of different variables on power demand, whether they are climatic or socioeconomic. Improving comprehension of the complex interactions between these variables and power demand becomes crucial for accurately predicting and managing power demand across different timescales. Accurate predictions of power demand can help energy providers to optimize generation and transmission, reduce costs, and improve the reliability of power supply at the seasonal scale. This becomes even more critical in the
context of climate change, which has already begun to impact power demand and caused power outages in various parts of the world due to high cooling demand associated with exceptional heatwaves and other extreme climate events (Ahmad, 2021; Burillo et al., 2018). Finally, going one step further, understanding the impact of climate change on power demand is essential for managing $CO_2$ emissions from the power sector, as it is closely related to the development of strategies for reducing greenhouse gas emissions and adapting to changes in energy
consumption patterns (Jiang et al., 2020).

While artificial intelligence techniques use has grown to investigate energy demand-side responses at various spatial and temporal scales (Antonopoulos et al., 2020), literature on the impact of climate and weather variability on power demand using these methods is still limited. Previous studies have primarily been developed for specific regions or countries (Mohammadiziazi and Bilec, 2020; Hiruta et al., 2022a; Hiruta et al., 2022b;
Gurriaran et al., 2022a; Gurriaran et al., 2022b). Until recently, there was no comprehensive worldwide dataset for daily power dynamics across multiple countries. This knowledge gap has been filled with the introduction of the Carbon Monitor Power data (Zhu et al., 2023), which provides daily estimates of power demand at the national level for about forty countries, along with detailed sources of supply. In this study, we use this newly available dataset to develop a machine-learning approach for modeling daily power demand by combining



climate variables and human activity indices, considering the impact of climate through cooling and heating demand proxies. In addition, we consider human activity indices, such as working days, weekends, and holidays, as well as the level of stringency of COVID-19 measures, which play a crucial role in determining power demand as they reflect the level of economic activity (Antoniadis et al., 2022; Hiruta et al., 2022b).

Building on our earlier work on Qatar and Japan (Gurriaran et al. 2022a, 2022b), the present study aims to

develop data-driven models that simulate daily power demand for a large number of countries with contrasted climates based on the Carbon Monitor Power demand dataset, and a comprehensive set of daily climate variables and human activity indices. Additionally, the study aims to infer the most important variables for each country or region and discuss differences that may arise between the countries. The data we used include total daily power production at a national or regional scale from 15 February 2019 to 15 October 2022, climate variables, and

human activity indices to develop models at a national or regional scale. Our study assumes that daily power production is equal to power demand, as transmission losses are assumed to be negligible. The dataset is divided into a learning set and a test set. Different machine-learning regressors are trained on the learning set to develop the models for power demand prediction. The performance of the models is assessed using the test set through error metrics, the evaluation of overfitting, and an analysis of the model's residuals.

The models developed in this study have the potential to be applied in various contexts. The same models could be used to define new responsive power production modules coupled with weather forecast models to enable operational production forecasting. They could also benefit the domain of air quality monitoring; for example, the models could be integrated into data assimilation systems of atmospheric composition, such as the global Copernicus Atmosphere Monitoring Service (CAMS) and regional models, which require interactive emissions

fields with weather and human activity variations. Furthermore, our models may be used for adapting power systems to climate extremes. Finally, they may be incorporated into longer-term climate scenarios assuming that the short-term climate response of power production will remain unchanged. Some of our models can even integrate hypotheses relative to changes in consumption habits.

We present models for eight countries or groups of countries. Those countries represent diverse climates,

economies, and populations worldwide: Australia, Brazil, China, the European Union (including the United Kingdom, referred to as EU27 & UK), India, Russia, South Africa, and the United States. Those countries are all significant in terms of population, GDP, power production, and $CO_2$ emissions. Together they represent about 50 % of the world's total population, 67 % of the global GDP, and 80 % of total power generation in 2021 (IEA, 2021). For the sake of presentation, we present the results for EU27 & UK in the main text as an illustration. The

results for other countries are provided in the Supplementary Materials (Sect. 2).



**Figure 1. Methodological flowchart of this study: CMP-SIM approach.**

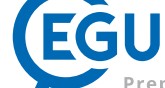

## 2 Data

This section describes the input data used to develop the regional or national models simulating power demand:
the Carbon Monitor Power – Simulators (CMP-SIM v1.0). Regional power demand refers here to the power
demand of EU27 & UK. All the data are at a national or regional level and at a daily timescale. The data were
pre-processed to a format suitable for the machine learning approach. We used 32 months of input data from 15
February 2020 to 15 October 2022.

**Predictive Features**

| | Variable Name | Unit | Description | Country/Region | Source |
|---|---|---|---|---|---|
| Climate Variable | T2M | °C | Average daily surface air temperature at 2m | All | ERA5 |
| | T2Mmax | °C | Maximum daily surface air temperature at 2m | | |
| | T2Mmin | °C | Minimum daily surface air temperature at 2m | | |
| | Td | °C | Average daily dew point temperature at 2m | | |
| | RH | % | Average daily relative humidity | | |
| | Surface Pressure | Pa | Average daily pressure of the atmosphere on the surface of the land | | |
| | U | m.s$^{-1}$ | Average wind speed and direction at 10m | | |
| | TP | m | Average daily total precipitation | | |
| | SSRD | J.m$^{-2}$ | Surface solar radiation downward | | |
| | STRD | J.m$^{-2}$ | Surface thermal radiation downward | | |
| Human Activities Indices | DOW | - | Day of week – categorical variable from 0 to 6 | All | Python repository |
| | Holidays | - | Categorical variable 0 or 1 | All but EU | Manually collected |
| | Workplace | % | Changes of workplace occupancy compared to a baseline | All but China and EU | Google Community Mobility Reports |
| | Covid | - | COVID-19 stringency index | All | Mathieu et al., 2020 |



| | Variable Name | Unit | Description | Country/Region | Source |
|---|---|---|---|---|---|
| | TOY | - | Numerical day of year | China and EU | - |
| | GDP | % | Quarterly GDP growth rate | Only China | China Bureau of Statistic |
| **Target Feature** | | | | | |
| | Variable Name | Unit | Description | Country/Region | Source |
| Power Data | Total Demand | GWh | Total daily power demand in the region considered | All | Carbon Monitor - Power |

**Table 1. Input and output dataset for this study**

## 2.1 Predictive features

The predictive features used to build models predicting power demand, including climate variables and human activity indices (Table 1), are described in the following.

Climate Variables: The climate variables include temperature (daily average, max, and min), dew point temperature, surface pressure, relative humidity, wind, precipitation, and solar radiation. These variables are known to impact power demand, as they affect the energy consumption patterns of households and industries. The climate variables are obtained from the ERA5 reanalysis at a daily timescale (Muñoz Sabater, 2019). All the climate variables were weighted by population density (CIESIN) to give more importance to climate over densely populated areas, as these regions are accountable for a significant proportion of power demand.

Human Activity Indices: Human activity data (Fig. 2), such as working days, holidays, and school vacations, also play a crucial role in determining power demand, as they reflect the level of activity influencing the power demand. These indices are obtained from publicly available datasets.

The effect of working days and weekends on power demand is accounted for with the numerical variable DOW (Day Of Week), where the value zero corresponds to Monday, one to Tuesday, and so forth, with six representing Sunday. To account for the effect of holidays, we introduce the variable "Holiday", which takes the value of one if the day is a holiday and zero otherwise.

Because our data cover the COVID-19 period, we accounted for the impacts of COVID-19-related measures on power demand using the COVID stringency index. We used the COVID stringency index, which aggregates information from various policy sources, including the Oxford COVID-19 Government Response Tracker (Hale et al., 2021) and the ACAPS COVID-19 Government Measures Dataset (ACAPS COVID-19). The COVID



stringency index is a composite measure comprising nine response indicators, such as school closures, workplace closures, and travel bans. The values of these indicators are rescaled on a scale of 0 to 100, where 100 represents the strictest level of response. The COVID stringency index is available for 207 countries (Mathieu et al., 2020). Additionally, we used data from the Google Community Mobility Reports to account for the effect of vacations on power demand. Google developed these reports to track the effects of COVID-19 on the frequency of various

types of locations and was available from 15 February 2020 to 15 October 2022 (Google LLC., 2020). These reports are constructed by analyzing location data from users who have opted into Location History for their Google account, and the data are aggregated to preserve users' anonymity. The reports indicate how visits and length of stay in these different location categories have changed over time compared to a baseline period before the COVID-19 pandemic from 3 January 2020 to 6 February 2020. Specifically, we used the "workplaces"

metric, which reflects the change in the percentage of people present at their workplaces compared to the baseline reference period. To remove the effects of weekends and holidays in the workplace metric, we applied a running mean on a 7-day basis and replaced the values of the holiday days to match the value of the previous day. This was done because the effects of weekends and holidays are already represented by the variables "DOW" and "holidays," respectively. However, the Google Community Mobility Reports data are unavailable for China and

the EU (Table 1). Thus, we employed an alternative variable, namely "Time of the Year" (TOY), to reflect the level of economic activity in the two countries. This variable is defined as the numerical day of the year, ranging from one on January 1st to 365 or 366 on December 31st. TOY is an alternative to Google Mobility data because it can serve as a proxy for economic activity by allowing the possible seasonal variation of power demand throughout the year to be linked to a specific period within that year.

Finally, given the significant reliance of China's power demand on its industrial sector, it is imperative to consider economic indicators that reflect changes in industrial activity. We hypothesized a strong relationship between GDP and industrial activity and assumed that the fluctuations in GDP could be used as a proxy for changes in industrial activity. Consequently, we added quarterly GDP as a predictive feature for China.

In total, 15 predictive features were used to simulate daily power demand. However, the exact number and

combination of predictive features used to simulate power demand vary depending on the availability of human activity data for a particular country and the use of GDP.

## 2.2 Target features

The target feature of this study, i.e., the data we aim to simulate, are the total daily power demand at the regional or national scale (Fig. 2). This feature is calculated from the publicly available Carbon Monitor - Power dataset





(Zhu et al., 2023; Liu et al., 2020a; Liu et al., 2020b). This dataset includes daily historical data on electricity generation from 37 countries since January 2019. It gives the electricity generated by different energy sources: fossil (coal, gas, and oil), renewable (solar, wind, hydro, and others including biomass, geothermal, etc.), and nuclear. We obtain the total daily power demand by summing the daily power generation of each source under the assumption that demand is equal to generation. One outlier was detected for India (19 April 2020) and

removed from the dataset.

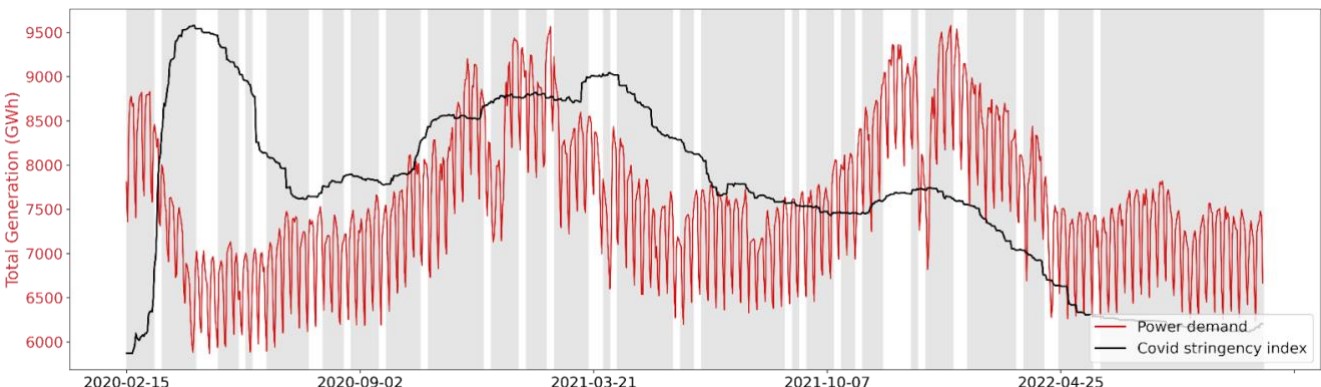

**Figure 2. Evolution of human activity predictive feature from COVID stringency index for EU27 & UK and power demand over the studied period. The shaded area represents the learning periods, and the**

**blank area the test periods.**

## 3 Model Development

This section describes the approach we developed for establishing national or regional models simulating daily power demand from the predictive features described in Sect. 2.1 (Fig. 1). The models have been coded in Python version 3.6.12. Our approach follows machine-learning procedures (Raschka, 2018; Raschka and Mirjalili, 2019),

including the formation of learning and test subsets, random search with cross-validation, regressor training on the learning set and performance evaluation with error metrics on the test set, model interpretation with model agnostic interpretability methods (ALE plots and permutation feature importance), and validation curve analysis to detect potential overfitting or underfitting. Previous studies have applied similar approaches to various countries. For example, in Japan, Hiruta et al. (2022a) used a machine-learning approach to derive temperature

response functions at an hourly timescale. In previous works, we have developed data-driven models for long-term predictions in specific regions, namely Qatar (Gurriaran et al., 2022a) and Japan (Gurriaran et al., 2022b). The approach developed for Japan was more detailed and tailored to the country. It included a separate model for



carbon intensity and was conducted on the Japanese regional scale. The approach presented in this study is more generic and can be applied to any country or region worldwide so long as daily data are available.

## 3.1 Partitioning of input data into learning and test subsets

We followed a consistent procedure for each country or region, as illustrated in Fig. 1. The first step is to divide the input dataset into learning and test subsets. This is a necessary step to examine the robustness of the results; machine-learning regressors will be trained on the learning set, and the performances of the models will be evaluated on the test set. The entire dataset is divided into blocks of one-week size. Then, all these blocks are shuffled randomly. Once the shuffling is done, 25 % of the data are assigned to the testing subset and 75 % to the learning subset (Fig. 2). This ratio is common for partitioning the dataset into learning and test subsets (Raschka, 2020; Raschka and Mirjalili, 2022). This process ensures that both subsets are representative of the whole dataset and that the results obtained are robust and reliable.

## 3.2. Random search with cross-validation and evaluating metrics

We evaluate four machine-learning regressors: Random Forest (RF) (Breiman, 2001), Gradient Boosting (GB) (Fisher, 1958; Chen and Guestrin, 2016; Ke et al., 2017), Multivariate Adaptive Regressions Splines (MARS) (Friedman, 1991), and Generalized Additive Model (GAM) (Hastie and Tibshirani, 1990). RF and GB are two ensemble learning methods. RF combines multiple decision trees to make more accurate predictions. Each decision tree is trained on a random subset of the training data to reduce the risk of overfitting. When making a prediction, RF takes the average prediction of all the decision trees in the ensemble. GB combines weak learners to form a stronger predictor. Each weak learner is trained sequentially to minimize the errors of the previous weak learners. This process is repeated until the error is minimized or a specified number of weak learners is reached. The final prediction is made by combining the predictions of all the weak learners. MARS and GAM are two interpreted machine-learning methods for regression analysis. MARS uses a sum of piecewise linear regressions to model non-linear relationships, while GAM uses a sum of smooth functions such as splines. For GAM, we specified an equation for each country using the backward feature selection process (Wood, 2017). The model was executed using all the predictive features; then, we gradually eliminated all non-significant features until the model's stability was achieved according to a Fisher test. The allocation of a specific number of splines to each feature was accomplished using an integer value approximately ⅓ higher than the degree of freedom estimated by the GAM regressor during the initial run.





All regressors are trained on the learning set, and their hyperparameters are optimized through a random-search process with 5-fold cross-validation on the same subset. The cross-validation process involves partitioning the learning data into multiple subsets (here 5) and uses each subset in turn as a validation set to assess the model's performance. The final evaluation of the model is done on the test set. The hyperparameters are the settings of the
regressors that need to be specified before the training phase. They are specific to the type of regressor used and cannot be learned from the data. Optimizing the values of the hyperparameters is important as they can impact the accuracy and performance of the models. Grid search and random search are two common techniques to tune hyperparameters. Grid search exhaustively searches through all possible combinations of hyperparameters, while random search randomly samples hyperparameters from a specified distribution. In a random search, the number
of combinations tried is controlled by a pre-determined number of iterations (n_iter). The high computational cost of grid search led us to choose random search to explore the hyperparameter space for RF, GB, and MARS. Limiting the number of iterations to 200 considerably reduced the computation time while giving satisfying results. For GAM, we optimize only two hyperparameters. The description of the hyperparameters optimized through the random search process can be found in the supplementary materials (Sect. 1).

We calculated various error metrics to evaluate the performance of the models on the test set. These include the coefficient of determination ($R^2$), Mean Absolute Error (MAE), Root Mean Squared Error (RMSE), and Median Absolute Error (MedAE). The objective is to maximize $R^2$ and minimize the values of MAE, RMSE, and MedAE.

**3.3. Interpretation of the models with permutation feature importance and ALE plots**

Permutation feature importance and ALE plots are two methods that allow the interpretation of non-directly interpreted machine learning models such as RF and GB regressors. We use the permutation feature importance to classify the predictive feature by order of importance and the ALE plot to understand the relationship between the predictive features and the target feature. For consistency in our results, we also apply this method to GAM and MARS regressors even though they are interpreted machine learning models.

Permutation feature importance enables a relative classification of features within the models, identifying the most significant predictive features to explain power demand for a particular country. We calculate a permutation score for each predictive feature with the four machine-learning regressors tested. This score is determined by randomly shuffling the feature and measuring the reduction in model accuracy that results. The feature is shuffled five times; then, an average score is calculated.





ALE plots enable interpreting the relationship between the target feature (power demand) and one particular predictive feature (Apley and Zhu, 2020). They represent the influence of the predictive feature on the target feature when the other predictive features are held constant. ALE plots are used to identify the non-linearities between the target feature and predictive features. In this study, ALE plots were calculated for all predictive features included in the model that achieved the best evaluation metrics. This calculation involves dividing the

range of the feature into intervals, calculating the average power demand for each interval, determining the differences in prediction between adjacent intervals, and integrating to estimate the individual influence of a feature.

### 3.4. Model validation: validation curves, quantile-quantile diagrams, autocorrelation, and seasonal decomposition

Validation curves are commonly used to detect overfitting or underfitting problems. Overfitting happens when a model is too complex and fitted to the training data to the point that the model cannot be generalized to other data. In this case, the model performs well on the train sets but poorly on the validation set. Two validation curves are calculated, one for the train set and one for the validation set to detect overfitting. Those curves show how the model's performance (here measured with $R^2$) changes for both subsets as a particular hyperparameter

value of the model is varied. If the model performs much better on the train set than on the test set or the two curves diverge above a certain hyperparameter value, it indicates overfitting issues. In this study, we also used the validation curves to verify that the correct hyperparameter values were selected during the random search process. Underfitting is detected when the performances of the model are poor on both subsets. The validation curves for all countries considered can be found in Supplementary Materials (Sect. S2).

Assumptions underpinning statistical methodologies are critical for ensuring the validity of analyses. One of our methodological assumptions is the normality of the residuals obtained from power demand calculations using our statistical models.  To verify this assumption, we constructed quantile-quantile (QQ) plots for the residuals obtained from the four regressors (Chambers, 1983). These plots display the quantiles of a dataset as a function of the corresponding theoretical quantiles of a normal distribution. If the points on the QQ plot align closely with

the diagonal, it indicates that the residuals follow a normal distribution, supporting the suitability of our methodology for accurately simulating power demand from the given data.

Assessing the temporal structure of the residuals of a model is another way to evaluate the validity of a time-series model. Autocorrelation plots represent the correlation between a time-series and its delayed version. We constructed autocorrelation plots for the residuals of our four power demand regression models to identify any





remaining temporal structures that the models may not have captured. To ensure the inclusion of weekly information, we chose a maximum time lag of 14 days for our analysis. If the autocorrelation values decrease rapidly as the lag increases, it suggests that our models have fully explained the temporal information. Conversely, if the autocorrelation values remain high at larger lags, some relevant temporal information may not have been captured.

Finally, we used time-series seasonal decomposition to assess the performances of our models at different time scales. Seasonal decomposition is a statistical technique that decomposes a time-series into different components: trend, seasonality, and residuals (Hyndman and Athanasopoulos, 2018). The trend component represents the long-term trend of the data, and the seasonality component captures the periodicity in the data (i.e., weekly, seasonal, or annual cycles). The residuals component is the random variations in the data that cannot be explained by the seasonal decomposition method. For this study, we used a simple decomposition method based on moving average with an additive model: $PD_t = T_t + S_t + R_t$, where $PD_t$ is the power demand time-series, $T_t$ is the "trend" component, $S_t$ is the "seasonality" (here weekly) component, and $R_t$ is the residual component. $T_t$ is estimated using a convolutional filter and then subtracted from $PD_t$. $S_t$ is obtained by averaging the de-trended series for each period. In this study, we did this analysis with a seven-day period to capture the weekly seasonality. This seasonal decomposition method was applied to the four time-series obtained with our models and to the original power demand time-series to serve as a point of comparison.

## 4 Results: Output of the models

This section presents the main outputs of our machine-learning approach with a focus on EU27 & UK: the performance of the different models tested, the permutation feature importance, and the ALE plots. The results for other countries can be found in the Supplementary Materials (Sect. 2).

Scatter plots show the modeled vs. observed power demand, with corresponding error metrics displayed on each subplot (Fig. 3). In the case of the EU27 & UK, all regressors perform similarly in evaluating metrics. $R^2$ Table 2 provides a summary of the evaluation metrics for all countries.







**Figure 3. Comparison of machine-learning regressors performance for EU27 & UK. Predicted power demand plotted against observed power demand for the four machine-learning regressors tested: (a) RF, (b) GB, (c) MARS, and (d) GAM. The red dashed line represents the 1:1 line of perfect agreement between predictions and observations.**

Comparing the results of all countries, the models perform best in predicting power demand for Russia, with an $R^2$ of 0.98. In contrast, they exhibit the poorest performance for China, with an $R^2$ always under 0.8 (Table 2).



The results presented in Table 2 do not reveal a single best regressor that consistently outperforms others across all countries.


| | | Australia | Brazil | China | EU27 & UK | India | Russia | South Africa | United States |
|---|---|---|---|---|---|---|---|---|---|
| Random Forest | $R^2$ | 0.80 | 0.87 | 0.75 | 0.92 | 0.85 | 0.96 | 0.81 | 0.89 |
| | MAE | 12.7 | 41.5 | 893.4 | 201.6 | 106.9 | 53.5 | 13.7 | 306.2 |
| | RMSE | 16.7 | 59.8 | 1161.9 | 258.7 | 149.1 | 72.0 | 18.7 | 394.1 |
| | MedAE | 10.2 | 29.1 | 732.1 | 178.7 | 76.3 | 38.6 | 10.8 | 250.5 |
| Gradient Boosting | $R^2$ | 0.84 | 0.93 | 0.77 | 0.93 | 0.83 | 0.98 | 0.77 | 0.91 |
| | MAE | 10.9 | 33.6 | 872.5 | 177.2 | 112.2 | 44.6 | 15.0 | 280.1 |
| | RMSE | 14.6 | 42.4 | 1128.6 | 236.9 | 159.3 | 58.7 | 20.6 | 361.4 |
| | MedAE | 8.4 | 28.4 | 666.3 | 135.8 | 78.5 | 35.5 | 11.7 | 216.9 |
| MARS | $R^2$ | 0.83 | 0.92 | 0.78 | 0.94 | 0.80 | 0.97 | 0.83 | 0.90 |
| | MAE | 11.5 | 36.2 | 810.3 | 163.4 | 121.8 | 50.3 | 13.02 | 309.2 |
| | RMSE | 15.1 | 46.5 | 1106.6 | 225.4 | 173.2 | 66.8 | 17.8 | 388.0 |
| | MedAE | 10.3 | 28.4 | 635.9 | 118.9 | 80.1 | 42.3 | 10.4 | 248.3 |
| GAM | $R^2$ | 0.81 | 0.92 | 0.77 | 0.93 | 0.81 | 0.97 | 0.83 | 0.94 |
| | MAE | 11.7 | 37.1 | 889.8 | 178.0 | 127.5 | 46.9 | 13.7 | 22.1 |
| | RMSE | 16.2 | 45.6 | 1120.1 | 245.5 | 171.6 | 64.1 | 17.9 | 292.7 |
| | MedAE | 9.0 | 33.1 | 753.9 | 135.6 | 105.6 | 34.9 | 11.4 | 169.3 |

**Table 2. Comparison of the performances of the four machine-learning regressors for the countries studied with the four metrics: $R^2$, MAE, RMSE, and MedAE.**

Figure 4 shows the five most important predictive features, as determined by the permutation feature importance. When focusing only on predictive climate features (pink in Fig. 4), all regressors recognize temperature as a significant predictor, featuring it within the top five variables. However, the specific temperature-related feature that emerges as significant differs among the models (T2M, T2Mmax, T2Mmin, or Td). Furthermore, the variable SSRD (solar radiation) consistently appears as a crucial predictor, as it is included within the top five predictors for all regressors except MARS. These results underscore the crucial role of climate-related features in predicting power demand. On the other hand, the analysis also highlights the relevance of human activity



features, with DOW, Covid, and TOY (blue in Fig. 4) always among the top five predictors. It is noteworthy that

the order of the top five predictors varies across different models.

Figure 4. Permutation feature importances of the five most important predictive features from four different machine-learning regressors for EU27 & UK: (a) RF, (b) GB, (c) MARS, and (d) GAM.



ALE plots were generated for the top five predictive features with the MARS regressor, which performed best for EU27 & UK (Fig. 5). The ALE plots confirm the strong impact of temperature-related predictors on power demand, with Td, T2Mmax, and T2Mmin being particularly influential. ALE plots for Td demonstrate a positive correlation between power demand and heating (when the temperature is decreasing), and ALE plots for T2Mmax have a positive correlation for cooling (when the temperature is increasing) requirements. ALE plot for

T2Mmin shows both effects. Examining the ALE plot for DOW (day of the week) reveals that power demand holds less significance during weekends than on weekdays. Finally, the ALE plot for TOY shows a decrease in power demand at the end and beginning of the year, corresponding to the holiday season. Overall, our findings here illustrated for EU27 & UK suggest that both climate and human activity factors are crucial in predicting power demand, and a comprehensive approach that considers both these aspects is needed to yield more accurate

results.





**Figure 5. ALE plots of the effect of the top-five predictive features from MARS for EU27 & UK: (a) T2Mmax, (b) T2Mmin, (c) TOY, (d) Td, and (e) DOW, where size represents the number of days in each category. Each ALE plot shows the partial dependence of the target feature on a predictive feature while keeping all other features constant. The x-axis represents the values for each feature, and the y-axis represents the corresponding change in the predicted value of the target feature.**






**Figure 6. Weekly seasonal decomposition analysis of the daily power demand data from four different models (RF, GB, MARS, GAM) as well as the observed data (Obs) for EU27 & UK: (a) Observed and modeled daily times series, (b) trend component, (c) weekly component, and (d) residual component. The legend in panel d represents the Pearson correlation coefficient between the models' residues and the observations' residues. The shaded area in the plots represents the maximum daily temperature (T2Mmax) in a 7-day running mean.**



The comparison of the modeled decomposed times series and the observed decomposed time-series enable
assessment of the ability of models to capture the diverse temporal patterns inherent in the data. By decomposing
the time-series generated by the models and comparing them with observed electricity demand, it becomes
possible to evaluate the models' ability to accurately replicate the various temporal patterns evident in the
observational data. Figure 6 focuses on December 2021 to illustrate the negative impact of the Christmas
holidays on power demand. Electricity demand remains low at the end of the month, possibly due to the high
temperatures observed during this period. The trend component (Fig. 6b) indicates that all models successfully
capture the decrease in power demand attributed to the Christmas break. Upon comparing the seasonal
decomposition of the models with that of the observational data, it demonstrates that GB exhibits the highest
accuracy in simulating this decrease in power demand. Additionally, our analysis demonstrates that all models
perform well in simulating the weekly component (Fig. 6c). Lastly, our investigation reveals a correlation
between the residuals of the seasonal decomposition of the models and those of the observations. This finding
suggests that the models effectively capture short-term temporal patterns in electricity demand, indicating their
potential to be used for generalization.



# 5 Discussion

## 5.1 Model inter-comparison in different countries




**Figure 7. Taylor diagram for simulated power demand for the eight countries or regions. The colors indicate the different regressors tested: green, RF; orange, GB; blue, MARS; purple, GAM. The radial**



**axis indicates the standard deviation, the angular axis the coefficient of correlation (R), and the dashed**
**circles the RMSE.**

To compare the performance of our models against the (test) observation and across the eight countries or regions, we constructed Taylor diagrams for each country (Fig. 7). These diagrams provide a comprehensive
visualization of how well the models compare to the reference data for each country in terms of correlation, RMSE, and standard deviation (Taylor, 2001). The results from the Taylor diagrams confirm what was observed in the previous section with the evaluating metrics (Table 2). Specifically, the performance of each model is similar for a given country or region, while it differs across countries. The models exhibit the best correlation with observation for Russia (close to 0.99), closely followed by the United States, EU27 & UK, and Brazil, with
a correlation higher or very close to 0.95. For Australia, China, India, and South Africa, the correlation is around 0.90. Except for India, the models underestimate the standard deviation of daily power demand.

One of the objectives of our study is to identify the most influential features on power demand for each country or region and to investigate whether any similarities exist across the different countries. A comparison of feature importance for each country and model (Fig. 8) is conducted to achieve this objective. Our results suggest that
temperature-related features, including T2M, T2Mmax, T2Mmin, and Td, are always the primary climate drivers of power demand in all examined countries, indicating their significant influence on power demand across different regions. The other climate-related features included in this study do not appear to significantly drive power demand, except for SSRD, which slightly influences power demand for some countries in the RF, GB, and GAM models.

Regarding human activity predictors, we observed significant variations in their importance across different countries and machine-learning regressors. For instance, the DOW feature shows high importance in some countries while insignificant in others, similarly for workplaces activity from Google Mobility data. In general, the different models found the same features to be the most important, even though the value of the feature importance varies across models. Quarterly GDP is a crucial feature for predicting power demand in China.
Without quarterly GDP, the evaluating metrics were poor, leading us to conclude that the models for China were unexploitable for generalization. These results highlight the importance of considering economic indicators reflecting the importance of the industrial sector's share in the total power demand, such as quarterly GDP, when developing models for power demand forecasting in China.





Overall, Fig. 8 provides insights into the key factors influencing power demand across various countries,
highlighting again the crucial role of temperature-related features as a primary driver of power demand. The
observed variations in the importance of human activity predictors across different countries and machine-
learning regressors suggest the significance of accurately including region-specific characteristics and machine-
learning approaches in predicting power demand.





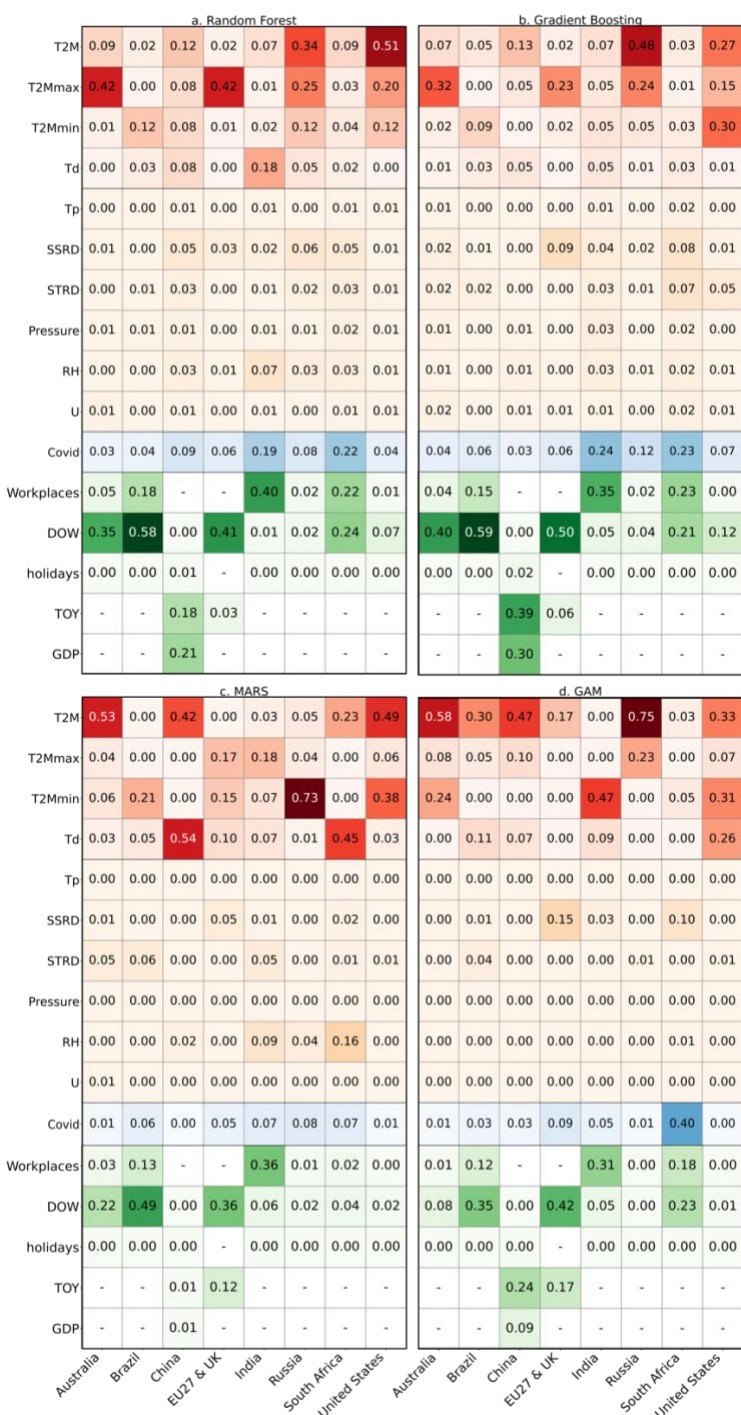

**Figure 8. Permutation feature importance for four different machine-learning regressors: (a) RF, (b) GB, (c) MARS, and (d) GAM. The colors represent the different types of predictive features: red for**



**temperature-related features, orange for other climate-related features, blue for covid, and green for socioeconomic features. The columns correspond to the countries indicated at the bottom of the figure.**

## 5.2 Validation and Limits of the Models

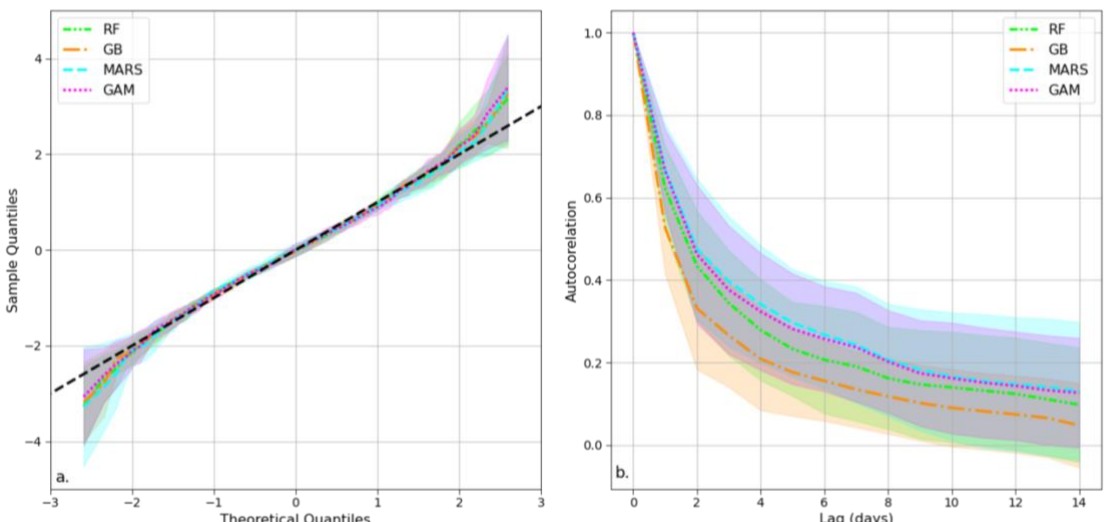

**Figure 9. (a) Quantile-quantile (QQ) plot displaying the mean and standard deviation (shaded area) of the residuals' quantiles for the four models (RF, GB, MARS, and GAM) across eight European countries during the test period. (b) Autocorrelation plot illustrating the average autocorrelation values across the eight countries or regions studied for each of the four models with a 14-day maximum lag. The shaded area represents the standard deviation across countries.**





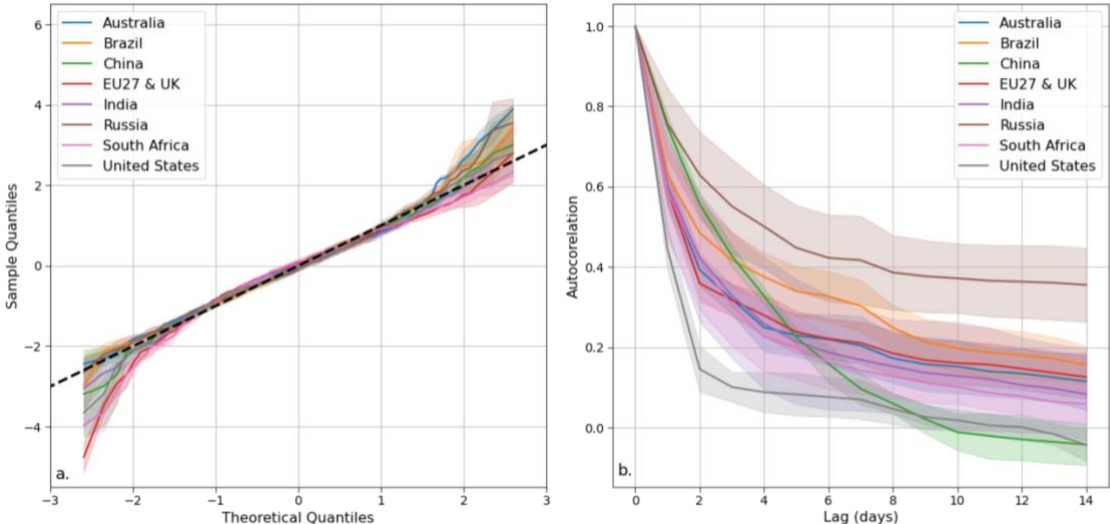

**Figure 10. (a) Quantile-quantile (QQ) plot displaying the mean and standard deviation (shaded area) of the residuals' quantiles for all countries across the four models during the test period. (b) Autocorrelation**
**plot illustrating the average autocorrelation values across the four models for all countries with a 14-day maximum lag. The shaded area represents the standard deviation across countries.**

The analysis of the residuals of the four models provides information on the performance of the models in predicting the statistical distribution of power demand. We analyzed the performance of the four models using residual quantiles compared to the theoretical Gaussian distribution (Figs. 9a and 10a). This examination is

carried out at a global level encompassing all countries and regions (Fig. 9a) and at a country-specific level for each model (Fig. 10a). This analysis reveals that all models perform similarly, with slight deviations from the expected normal distribution within the intermediate quantiles range (between two and minus two) and higher deviations observed above this threshold. Therefore, the Gaussian hypothesis is confirmed, except for extreme values, for which the dataset contains relatively few observations. Those extreme values are often attributed to

periods of unusual economic activity, such as bank holidays or specific public holidays that are difficult to model (Srinivasan et al., 1995; Ziel, 2018). Consequently, our models can underestimate or overestimate very low or high power demand, respectively.

The autocorrelation plots of the residuals (Fig. 9b) reveal differences between the models and countries. In

particular, gradient boosting outperforms the other models in this respect, with the lowest autocorrelation values. In contrast, MARS shows the highest values. RF and GAM are in between with very similar results. Some countries exhibit superior performances (Fig. 10b). For example, the residual autocorrelation values for Russia





decrease with time at a slower pace than for other countries. Despite the differences in autocorrelation values between the models and across countries, it is worth noting that all models exhibit a similar trend. Specifically, the autocorrelations of the residuals are high up to a lag of a few days, as also reported elsewhere. The autocorrelations drop beyond a lag of a few days, indicating that our models did not miss any significant temporal information.

Overall, these findings are encouraging and validate our models. Therefore, the models can be used for the projection of power demand. However, caution should be exercised when considering extreme values. It is possible to improve the modeling of such values by using a class of quantile regression models. Various types of models have been developed that are specifically designed to address extreme quantiles. One such model is the quantile regression forest, which is a generalization of the random forest model (Meinshausen, 2006). Another example is the additive quantile regression model, which has demonstrated promising results in recent studies (Fasiolo et al., 2020). Such models can be applied in future studies to improve the accuracy of power demand projections.

Overall, while the models developed in this study offer valuable insights into predicting power demand, some limitations must be considered. Firstly, our study period included the COVID-19 pandemic, which significantly impacted energy consumption and emissions (Liu et al., 2020b; García et al., 2021; Aruga et al., 2020). While we incorporate this variable in our models, the extent of its impact may not have been fully captured. Training the regressors on periods not affected by covid might give better results.

Additionally, the irregularity observed in the modeling process for China is worth noting, as China necessitates the inclusion of quarterly GDP to attain good results. Although our approach was largely consistent across countries, it did not achieve a perfect "one-fits-all" approach. Consequently, while this work established a modeling framework applicable to multiple countries, further revisions may be required when extending it to countries not encompassed in this study.

Finally, although all models yielded satisfactory outcomes, each model employed the predictive features in distinct ways (Figs. 4 and 9). Certain predictive features did not exhibit the expected behavior (as shown in Fig. 5b, where T2Mmin showed no sensitivity for lower temperatures). Furthermore, the role and impact of the TOY variable, which functions as a corrective factor for countries where Google Mobility data are not available, remain somewhat ambiguous. While it can account for annually recurring phenomena not elucidated by other predictive features, it would require a more extensive dataset spanning several years to refine its precise function. Addressing these limitations through future research can lead to more accurate and robust models for predicting power demand and related $CO_2$ emissions.



## 6 Perspective

**Figure 11. Extended methodological flowchart toward CO₂ emission projections.**

The present study aims to establish a modeling approach to simulating national daily power demand from climate and human activity features. The proposed approach has the potential to be extended to predict long-term power demand trends under changing climatic conditions and to estimate the corresponding $CO_2$ emissions resulting from power generation (Fig. 11). This extension would involve developing separate models for simulating power demand and carbon intensity. To achieve this, the target variable would be set as the daily carbon intensity rather



than power demand, resulting in the development of two parallel models: one for daily power demand and another for daily carbon intensity. $CO_2$ emissions are calculated by combining the projections from these two

models.

To apply this approach, projected climate features obtained from the CMIP6 simulation round, along with projected human activity variables such as DOW (Day of the Week) and Holidays, would be necessary. It should be noted that certain predictive features, such as "workplaces" from the Google Mobility Data, may not be subject to projection. By employing the two abovementioned models, projections of power demand, carbon

intensity, and $CO_2$ emissions can be obtained. Other socioeconomic factors, such as population growth, GDP, and environmental policies, can be incorporated to enhance the projection of daily power demand and $CO_2$ emissions (Figure 11). By considering the influence of population growth and GDP, the projections of daily power demand can be scaled accordingly. Carbon intensity projections could be developed based on assumed environmental policies aligned with the SSPs (Shared Socioeconomic Pathways) narratives and used to scale the projection of

daily carbon intensity obtained with the data-based models. A similar approach has already been applied to Qatar and Japan with different scenarios in alignment with the SSPs narratives (Gurriaran et al., 2022a; Gurriaran et al., 2022b).

The approach presented in this study has the potential to be extended to evaluate the effectiveness of different policies and initiatives aimed at reducing $CO_2$ emissions. By considering the influence of changing energy

demand under future climate change scenarios, it becomes possible to evaluate the effectiveness of these measures in achieving emission reduction targets. Furthermore, coupling the models developed in this study with simple climate models such as ACC2 (Aggregated Carbon cycle, atmospheric chemistry, and climate model, Tanaka et al., 2007; Tanaka and O'Neill, 2018) enables quantification of the feedback loop between human activity, $CO_2$ emissions, climate change, power demand changes, $CO_2$ emission changes, and the impact on

climate (precisely, human activity $\rightarrow$ $CO_2$ emissions $\rightarrow$ climate change $\rightarrow$ human activity).

In conclusion, the models developed in this study provide a valuable tool for analyzing, forecasting and understanding power demand patterns and $CO_2$ emissions in the context of climate change across various regions worldwide. Applying these models could offer insights into the potential future scenarios and dynamics of power demand, enabling policymakers and stakeholders to make informed decisions and shape effective energy policies.





**Code availability**

Access to the model's source code, stored in a private Zenodo repository, is available upon request at: https://doi.org/10.5281/zenodo.8135971. The model is coded in Python (version 3.6.12). The code is not publicly accessible as the primary company associated with the lead author has enforced a strict policy against its public distribution. The company's rationale for this decision is to safeguard its competitive advantage and proprietary algorithms from potential misuse or unauthorized access. The distribution of the code for non-commercial research purposes may be considered upon request to the corresponding author, subject to validation by the primary company. The reviewers were granted access to the code for evaluation purposes.

***List of Python library necessary:***

- Pyearth 0.1.0
- Pandas1.1.5
- Matplotlib3.3.2
- numpy1.19.2
- sklearn0.0
- pygam0.8.0
- PyALE1.1.2
- Statsmodels0.12.2

**Data availability**

Climate data used for this study are available from the global atmospheric reanalysis dataset produced by the European Centre for Medium-Range Weather Forecasts (ECMWF) at https://cds.climate.copernicus.eu/cdsapp#!/home. Below is the query necessary to download the data with the CDS toolbox and the list of the climate data names. This query allows you to access the specific variables you need by replacing "VARIABLE_NAME" with the name of the variable you want to download. Energy data are available at https://power.carbonmonitor.org (Carbon Monitor – Power). Google Community Mobility Reports data are available at https://www.google.com/covid19/mobility/. COVID-19 stringency index data are extracted from the Oxford Coronavirus Government Response Tracker (OxCGRT) project and are available at https://ourworldindata.org/covid-stringency-index.





ERA5 climate data names: 10m_u_component_of_wind, 10m_v_component_of_wind,

2m_dewpoint_temperature, 2m_temperature, relative_humidity, surface_pressure,

surface_solar_radiation_downwards, surface_thermal_radiation_downwards, total_precipitation

```
import cdstoolbox as ct

@ct.application(title='Download data')

@ct.output.download()

def download_application():

  data = ct.catalogue.retrieve(

    'reanalysis-era5-land',

    {

      'variable': 'VARIABLE_NAME',

      'year': '2022',

      'month': '01',

      'day': [

        '01', '02', '03', '04', '05', '06', '07', '08', '09', '10', '11', '12',

        '13', '14', '15', '16', '17', '18', '19', '20', '21', '22', '23', '24',

        '25', '26', '27', '28', '29', '30', '31',

      ],

      'time': [

        '00:00', '01:00', '02:00', '03:00', '04:00', '05:00', '06:00', '07:00', '08:00',

        '09:00', '10:00', '11:00', '12:00', '13:00', '14:00', '15:00', '16:00', '17:00',

        '18:00', '19:00', '20:00', '21:00', '22:00', '23:00',

      ],
```



```
    }

)

daily_mean = ct.cube.resample(data, freq='day', dim='time', how='mean')

return daily_mean
```

To obtain T2Mmax and T2Mmin, replace `how='mean',` by `how='max'` and `how='min',` repectively, in the penultimate line of the query

## Supplement

The supplement related to this article is available online at: https://doi.org/10.5281/zenodo.8039225

## Author contribution

LG led the design and implementation of the model with inputs from YG and collected the data. LG led the writing of the manuscript, with input from all authors. BZ constituted the Carbon Monitor – Power database. LG, KT, PC, and YG conceptualized the project and participated in the formal analysis of the results. All authors read and approved the manuscript.

## Competing interests

The authors declare that they have no conflict of interest.

## Acknowledgments

KT benefited from State assistance managed by the National Research Agency in France under the Programme d'Investissements d'Avenir under the reference ANR-19-MPGA-0008.

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
