# Peer review of "Carbon Monitor Power - Simulators (CMP-SIM v1.0) across countries: a data-driven approach to simulate daily power demand"

_EGUsphere, 2023_

## Author Response (AR1)

**Author's Response to the Reviewer's Comments**

**Reviewer Matteo de Felice:**

The article covers a very interesting topic and the authors try to assess a wide range of models in multiple regions. However, the article presents two main flaws:

1. The authors do not explain what the goal of this power demand model would be, given the challenges posed by this task, its target use is fundamental to understand the quality (and the usefulness) of the results

[Response]
We have clarified the purposes of our daily power demand models by adding the following paragraph in the introduction:
*"Our models have a multifaceted goal that includes separating climate factors that drive power demand variations, conducting cross-country comparative analysis of these factors, understanding the relationship between power and climate extremes, and developing predictive models for various applications. Our research aims to comprehensively understand the intricate interplay between climate and power demand by isolating specific climatic elements that influence energy consumption and comparing global variations. The main goal is to provide predictive models that can be used for different timeframes. This will help improve decision-making in energy management for short-term grid optimization, seasonal resource planning, and long-term strategies that align with changing climate scenarios."*

2. There is no mention to many works on the link between electricity demand and meteorological factors, as for example in the Copernicus Climate Change Service ECEM project and papers like (this is just an example)
https://rmets.onlinelibrary.wiley.com/doi/10.1002/met.1858,
https://iopscience.iop.org/article/10.1088/1748-9326/11/12/124025/meta,
https://iopscience.iop.org/article/10.1088/1748-9326/aa69c6/meta

[Response]
Thank you for suggesting additional references. While our initial literature review focused primarily on machine learning load forecasting, we recognize the importance of including references on the relationship between climate/weather variability and electricity demand. We expanded our literature review in the introduction and provided selected examples of studies investigating the climate sensitivity of power demand:
*"Studies have started addressing the aforementioned question from both power generation and demand perspectives. Examining the weather sensitivity of the power sector from a generation perspective can provide valuable insights for addressing these issues. In particular, several studies showed that increasing renewable generation capacity led to reduced baseload generation from fossil energy (Bloomfield et al., 2016, Silva et al., 2018), contributing positively to decarbonization. For example, increasing wind power generation in the UK reduced coal, gas, or nuclear power generation (Bloomfield et al., 2016). However, transition to renewables also increases the exposure of the power systems to climate variability (Craig et al., 2018, Elliston et al., 2013, Silva et al., 2018).*
*Other studies present approaches to identify meteorological, socioeconomic, and technical drivers for power demand and sectoral power production (Bloomfield et al., 2020, Toktarova et al., 2019). Such approaches provide improved means to quantify the impacts of climate change on the power system and are adaptable to different geographical locations. In addition, some studies focus on the development of databases that can be used for investigating the climate sensitivity of the power sector and the impacts of climate change, such as the C3S Energy database developed by Dubus et al. (2021), which provides power demand and power supply data for Europe."*

In addition, I would highlight a few issues that should be addressed:

1. The authors should use a simple model as a baseline (for example a linear regression) to show the value of using more complex methodologies to model the power demand. In other words, showing the added value of non-linearity or ensemble approaches.

[Response]
Based on the references added in the introduction, we argue that the existing evidence demonstrates the superior performance of GAM and machine learning approaches compared to more simple multi-linear models. This leads us to question the necessity of including a benchmark study in our article to show the added value of nonparametric approaches. Our objective is not to unequivocally endorse machine learning models but to assess their performance, identify limitations, and make comparisons among them. However, recognizing the importance of a comprehensive evaluation, we are open to including a benchmark section in another revised version of the article if you deem it necessary.

We also want to emphasize that our chosen approach, particularly with GAMs, is designed for flexibility and ease of inclusion of multiple variables. The interpretability of GAMs, represented as a sum of spline functions, allows for a straightforward understanding of each explanatory feature. For other machine learning models, we use interpretability methods to automatically identify significant variables. This, in turn, facilitates the exclusion of climatic factors that do not contribute meaningfully to our electricity demand forecasting studies. In addition, our approach minimizes the need for manual tuning compared to multi-linear models, providing efficiency and automation.

We added a paragraph in the introduction to clarify our point and provide a clearer overview of our study's contribution:
*"Electricity demand modeling often uses multi-linear models to integrate various influencing factors (Bloomfield et al., 2016 and 2020, Delort Ylla et al., 2023, Tantet et al., 2019, Toktarova et al., 2019). While such multi-linear models may appear more intuitive and simpler than machine learning models, they do not necessarily imply easier implementation and may require significant manual parameter tuning. Furthermore, machine learning models and semiparametric additive approaches, such as General Additive Models (GAM), are already widely used in the load forecasting community (Dordonnat et al., 2016, Fan and Hyndman, 2012, Nedellec et al., 2014, Obst et al., 2021, Pierrot and Goude, 2011) and have demonstrated superior forecasting capabilities compared to multilinear models (Hong et al., 2016)."*

2. The authors define the power demand as the total generation, assuming no cross-border exchanges of electricity that happen in many of the selected regions (e.g., exchanges between US and Canada). I think that defining this methodology as "daily power demand simulation" is a bit stretched, perhaps it would be more correct to change the title of the paper to "simulate daily power generation".

[Response]
We acknowledge your suggestion to change "daily power demand simulation" to "simulate daily power generation" and we incorporated this change in the revised version of the manuscript.

3. TOY is not correctly coded in the methodology, using a linear factor put 1st January and 31st December at the opposites, while they are actually consecutive. I would suggest using a sinusoidal function.

[Response]
We argue that the coding of TOY variables is correct for GAM models because it automatically generates cyclic splines. These cyclic bases have an additional constraint which requires continuity at the endpoints of the spline, making 1st January and 31 December close together. While sinusoidal functions could

improve accuracy for other machine learning models, our current approach with tree-based models efficiently handles these variables thanks to its ability to easily generate thresholds, thus allowing the first and last days of the year to be placed in the same category.

**Reviewer Giacomo Falcetta:**

Many thanks for the opportunity to review this very interesting, well written, and comprehensively presented paper.

While align with all the comments/criticism pointed out by Referee 1, in particular on the necessity of revising the terminology (e.g., generation, and not demand) and of testing simpler models to show the value added of non-parametric statistical modelling, I have a couple of additional comments to add.

First of all, while certainly significant and novel, the study should cite previous similar papers, e.g. (just an example, there is likely more) https://www.sciencedirect.com/science/article/pii/S0142061518336196, which are missing from the review of the literature in the first part of the paper. The authors should then better emphasize their contribution compared to previous large-scale energy production/generation demand studies.

[Response]
We agree with your comment, and, as mentioned in our response to the previous comment from the first reviewer, we improved the literature review in the Introduction to better contextualize our work and highlight its contribution by adding those paragraphs:
*"Studies have started addressing the aforementioned question from both power generation and demand perspectives. Examining the weather sensitivity of the power sector from a generation perspective can provide valuable insights for addressing these issues. In particular, several studies showed that increasing renewable generation capacity led to reduced baseload generation from fossil energy (Bloomfield et al., 2016, Silva et al., 2018), contributing positively to decarbonization. For example, increasing wind power generation in the UK reduced coal, gas, or nuclear power generation (Bloomfield et al., 2016). However, transition to renewables also increases the exposure of the power systems to climate variability (Craig et al., 2018, Elliston et al., 2013, Silva et al., 2018).*
*Other studies present approaches to identify meteorological, socioeconomic, and technical drivers for power demand and sectoral power production (Bloomfield et al., 2020, Toktarova et al., 2019). Such approaches provide improved means to quantify the impacts of climate change on the power system and are adaptable to different geographical locations. In addition, some studies focus on the development of databases that can be used for investigating the climate sensitivity of the power sector and the impacts of climate change, such as the C3S Energy database developed by Dubus et al. (2021), which provides power demand and power supply data for Europe.*
*Electricity demand modeling often uses multi-linear models to integrate various influencing factors (Bloomfield et al., 2016 and 2020, Delort Ylla et al., 2023, Tantet et al., 2019, Toktarova et al., 2019). While such multi-linear models may appear more intuitive and simpler than machine learning models, they do not necessarily imply easier implementation and may require significant manual parameter tuning. Furthermore, machine learning models and semiparametric additive approaches, such as General Additive Models (GAM), are already widely used in the load forecasting community (Dordonnat et al., 2016, Fan and Hyndman, 2012, Nedellec et al., 2014, Obst et al., 2021, Pierrot and Goude, 2011) and have demonstrated superior forecasting capabilities compared to multilinear models (Hong et al., 2016)."*

Moreover, the authors validate the model using daily resolution power generation. I think a crucial and valuable addition to demonstrate the extent to which the model and output data can be used for planning purposes would be to also evaluate the model error in each country/region in terms of

weekly/monthly/seasonal peak. This is because the peak load (maximum value)'s magnitude and modelling error are of great importance if the data is used in future studies and/or for planning and policy support purposes.

[Response]
Thank you for your comment. In terms of evaluating the performance of our model under extreme conditions, we acknowledged its limitations in dealing with extremes in the discussion section. In our future studies, we plan to explore specialized models designed for extreme conditions, which is also mentioned in the discussion. We argue that it would be more pertinent to evaluate our models' peak predictions once those changes implemented. We slightly modified the paragraph from the discussion to clarify this point and to add additional references (the sentence we added or modified are highlighted in red):

*"Overall, these findings are encouraging and validate our models. Therefore, the models can be used for the projection of power demand. However, caution should be exercised when considering extreme values. It is possible to improve the modeling of such values by using a class of quantile regression models. Various types of models have been developed that are specifically designed to address extreme quantiles. One such model is the quantile regression forest, which is a generalization of the random forest model (Meinshausen, 2006). Recent work by Gnecco et al. (2023) also proposed an approach based on random forest, tailored to extreme quantiles. Another example is the additive quantile regression model, which has demonstrated promising results in recent studies (Fasiolo et al., 2020). Finally, Velthoen et al. (2022) developed similar quantile models but for a gradient boosting approach. Such models are consistent with the type of models used in this study and will be applied in future studies to improve the accuracy of power demand projections."*

Finally, and relatedly, it would be interesting if the authors could explicitly account for the availability and use of cooling and heating technologies in each country, as these are strongly affecting the relation between meteorological variables and energy consumption, see https://www.nature.com/articles/s41598-023-31469-z

[Response]
Thank you for this suggestion. We are currently working on multi-country modeling to address data gaps, particularly in regions with limited air conditioning infrastructure, such as Europe compared to Japan or the US. This involves taking climate-energy demand relationships from one country and applying them to another. For example, we plan to simulate European electricity demand using the electricity-demand-climate relationship observed in Japan above the cooling threshold. In addition, our ongoing project involves refining the projection aspect by incorporating finer spatial resolution data. This includes integrating more socio-economic predictors, including demographic characteristics and building characteristics such as insulation and exposure. We added a paragraph explaining that we need to account for the availability of cooling and heating systems in our future long-term projections in the Perspective section of our reviewed manuscript:

*"However, our current models lack the ability to account for the future availability of heating and cooling technologies in the different areas under study. To address this significant limitation in projecting long-term trends, our strategy is to apply relationships observed in countries currently equipped with such technologies to countries lacking them. For example, we can simulate European electricity demand by applying the observed electricity-demand-climate relationship in Japan or the US, especially beyond the cooling threshold. By carefully selecting country combinations, we aim to develop future scenarios that are consistent with the narratives of the SSPs."*